# Development and Psychometric Properties of a New Patient-Reported Outcome Instrument of Health-Related Quality of Life Specific to Patients with Gambling Disorder: The Gambling Quality of Life Scale (GQoLS)

**DOI:** 10.3390/ijerph191710806

**Published:** 2022-08-30

**Authors:** Nicolas A. Bonfils, Henri-Jean Aubin, Marie Grall-Bronnec, Julie Caillon, Pascal Perney, Frédéric Limosin, Amandine Luquiens

**Affiliations:** 1CESP, UVSQ, INSERM, Université Paris-Saclay, 94804 Villejuif, France; 2Department of Psychiatry and Addictology, Assistance Publique-Hopitaux de Paris, Hôpitaux Universitaires Paris Ouest, 92130 Issy-les-Moulineaux, France; 3Faculté de Médecine, Université Paris Descartes, Sorbonne Paris Cité, 75000 Paris, France; 4Inserm, U894, Centre Psychiatrie et Neurosciences, 75013 Paris, France; 5Assistance Publique-Hopitaux de Paris, Hôpitaux Universitaires Paris-Sud, 94804 Villejuif, France; 6Faculté de Médecine Paris Sud, Université Paris XI, 91405 Orsay, France; 7Addictology and Psychiatry Department, CHU Nantes, 44093 Nantes, France; 8Inserm, U1246, Université de Nantes, Université de Tours, 44093 Nantes, France; 9Department of Addictions, CHU Nîmes, Université de Montpellier, 30000 Nîmes, France

**Keywords:** quality of life, problem gambling, patient-reported outcome, psychometrics, health-related quality of life

## Abstract

Background. Impairment or distress caused by gambling disorder can be subjectively assessed via quality of life. The aim of this study was to develop a new patient-reported outcome instrument to explore the health-related quality of life (HRQoL) in gambling disorders, the Gambling quality-of-life scale (GQoLS), and to document its psychometric properties. Methods. A previous qualitative study had been conducted using focus groups of problem gamblers to identify areas of HRQoL impacted by gambling. The seven domains identified served as the basis for the hypothetical structure of GQoLS. Draft items were generated from the patient’s speeches to illustrate each of these domains. Cognitive debriefing interviews were realized to obtain a final hypothetical GQoLS. A validation study was then carried out to determine the final version of GQoLS and its psychometric properties (structural validity, construct validity, internal consistency). Results. The final GQoLS was composed of 21 items, with a total mean score of 38.3 (±13.6). Structural validity found a major dimension and four other minor dimensions. The five dimensions were: “emotion”, “lifestyle”, “loneliness”, “taboo” and “preoccupation”. GQoLS was moderately to strongly correlated with PGSI and EQ-5D visual analogic scale. Cronbach’s alpha coefficient was 0.92. Conclusion. GQoLS is the first HRQoL instrument specific to patients with a gambling disorder and developed from the patient’s perspective. GQoLS presents good psychometric properties. GQoLS can be used in clinical research to demonstrate the effectiveness of an intervention on outcomes that are relevant from the patient’s perspective.

## 1. Introduction

Problem gambling is characterized by persistent and recurrent gambling behavior leading to clinically significant impairment or distress [1]. Impairment or distress can be objectively assessed via diagnostic criteria or gambling-related harms, or subjectively via quality of life (QoL) [1,2]. QoL captures patients’ subjective feelings about domains of functioning that are important to them [3]. QoL represents a useful indicator because it is adapted to patients’ desire to go beyond simple symptom management [4], predicts treatment adherence [5] and corresponds to multiple areas of functioning at all stages of recovery [6].

Treatment outcomes remain poorly defined in the field of gambling [7], probably due to the lack of a consensual definition of recovery. The course of the disease can differ between subjects and over time [8,9,10]. The Banff Consensus provided a framework with the minimum features of reporting the efficacy of intervention and supported the relevance of QoL assessment in gambling disorders, in parallel with the reduction of gambling behavior and the reduction of problems caused by gambling [11]. 

In 1993, the World Health Organization defined QoL “as an individual’s perception of their position in life in the context of the culture and value systems in which they live and in relation to their goals, expectations, standards and concerns” [12]. QoL instruments either explore overall QoL independently of any health condition or health-related QoL (HRQoL), classically involving four areas: physical, well-being, psychological state and social relations [13]. Several HRQoL instruments are used in gambling research and can measure the effectiveness of an intervention [14]. The areas of life explored by QoL and HRQoL instruments used in the past in gambling disorders were relationships with others, activities, physical state, psychological state, financial concerns, medical care and satisfaction with life [14]. However, they might not evaluate specific areas of HRQoL impacted affected by gambling disorder, are not totally validated in the problem gambling population and are probably less sensitive than gambling behavioral criteria [14]. Finally, despite being self-administered and requesting subjective answers, these instruments do not necessarily explore the entire spectrum of patients’ concerns on the impact on HRQoL of gambling disorder and do not take into account the patient’s subjectivity [14]. It was shown in other addictions as stigmatized as gambling disorders that a patient’s contribution was relevant to developing HRQoL instruments, such as in alcohol use disorder and people who inject drugs, to capture specific addictive behavior areas, but also general areas not part of generic HRQoL instruments, such as self-esteem [15,16]. Our conception of HRQoL is then not QoL reduced by gambling, but all possible areas of life impacted by a gambling disorder in a comprehensive and subjective meaning. An improvement in HRQoL in this meaning shall not necessary overlap with symptom disappearance, but more with personal recovery.

HRQoL is a specific type of patient-reported outcome (PRO) that is defined as any report of the status of a patient’s health condition directly by the patient, without interpretation of the patient’s response by a clinician or anyone else. As a first step in developing a new patient-reported HRQoL outcome instrument specific to patients with gambling disorders, we conducted six focus groups with 25 current or lifetime at-risk problem gamblers to identify key domains of QoL impacted by problem gambling [17]. We identified seven key domains of the impact of problem gambling [15]: loneliness, financial pressure, relationship deterioration, feeling of incomprehension, preoccupation with gambling, negative emotions, and avoidance of helping relationships. These results support the relevance of developing a specific HRQoL scale in the context of gambling.

The aim of this study was to develop a hypothetical frame and draft a new PRO instrument to explore HRQoL in gambling disorders, the Gambling Quality of Life Scale (GQoLS), and to document its psychometric properties.

## 2. Methods

### 2.1. Development of the Hypothetical Framework of GQoLS

#### 2.1.1. Development of the Draft of GQoLS

Seven key areas of HRQoL affected by problem gambling were identified by Bonfils et al. through a content analysis of 6 focus groups in 25 current or lifetime at-risk problem gamblers, performed with Alceste^©^ software (ALCESTE, V. 2015, Image, Toulouse, France). The seven areas formed the basis of the hypothetical conceptual framework for the GQoLS: loneliness, financial pressure, relationship deterioration, feeling of incomprehension, preoccupation with gambling, negative emotions, and avoidance of helping relationships [17].

#### 2.1.2. Generation of Items of the GQoLS

Draft items were generated to illustrate each of these areas. NB, and AL, psychiatrists specialized in gambling addiction, generated 94 items using the significant verbatim from the corpus of our previous study [17]. The items were generated using directly the verbatim of participants from the first study, as unprocessed as possible. Some items could then overlap, but we gave particular attention to generating several items for each of the 7 areas. We chose a 4-point-scale (“not at all”, “a little”, “quite a lot”, and “very much”, respectively, scoring 0, 1, 2, 3) to balance responder burden and a 4-week recall period [16].

#### 2.1.3. Expert Review

The initial 94 items were reviewed by psychiatrists and addictologists (AL, HJA and NB) and reduced to 87 by removing overlapping items, items expressing multiple concepts, and items overly specific to a particular individual context.

#### 2.1.4. Cognitive Debriefing Interviews

To assess the face and content validity of GQoLS, subjects were recruited from two sites in France: (a) one addiction specialized treatment service in French hospital: Paul-Brousse hospital in Villejuif; (b) Gamblers Anonymous meetings in Paris. Subjects with a current gambling disorder or in remission were invited by face-to-face or telephone contact to enroll. Inclusion criteria were: (i) a lifetime or previous 12 months Canadian Problem Gambling Index-Problem Gambling Severity Index (CPGI-PGSI) score > 3; (ii) age ≥ 18; (iii) subjects had to give their signed and informed consent and had to be affiliated to Social Security. Exclusion criteria were: (i) learning difficulties that prevented reading and responding to questionnaires; (ii) major physical comorbidity as judged by the investigator to have a significant influence on the subject’s day-to-day life; (iii) major psychiatric comorbidity as judged by the investigator to have a significant influence on the subject’s day-to-day life (e.g., current manic or current major depressive episode); (iv) current addictive comorbidity as assessed by clinicians’ judgment (with the exception of nicotine dependence); (v) significant cognitive impairment, as judged by the investigator; (vi) being under curatorship or guardianship. Eleven participants (*n* = 8 in Villejuif hospital and *n* = 3 in the Gambler Anonymous meeting) participated in the cognitive debriefing interviews: six in the first round of cognitive debriefing interviews and five in the second round. Demographic and clinical characteristics of the participants are presented in Table 1.

During the first round of cognitive debriefing interviews, participants found the questionnaire easy to understand and complete. Participants reported that most items were relevant to their experience with gambling. Some changes were made: 37 items considered ambiguous, redundant, difficult to understand or extreme were removed; and 10 items were revised for clarity. Most participants reported that 4-week recall period was too short as gambling practice was not necessarily continuous. Thus, a 6-week recall period was proposed for the second round of interviews. During the second round of cognitive debriefing interviews, participants found the revised questionnaire easy to understand and complete, and they considered the items and examples to be relevant to their experience with gambling. Participants took less than ten minutes to complete the questionnaire. Minor revisions were made to one item and to the instructions for more clarity. The longer 6-week recall period was deemed suitable. The final draft of the GQoLS was composed of 50 items and 7 a priori dimensions hypothesized during the development stage: loneliness (items 1–8), financial pressure (items 9–21), relationships deterioration (items 22–29), preoccupation with gambling (items 30–36), feeling of incomprehension (items 37–38), negative emotion (items 39–47) and avoidance of helping relationships (items 48–50).

### 2.2. Psychometric Properties of GQoLS

#### 2.2.1. Subjects and Setting

This study is part of the TRAIN-ONLINE study, a comparative, randomized controlled trial with no face-to-face, taking place in France (Clinical trial reference: NCT03673800). TRAIN-ONLINE aims to assess the clinical efficacy of an online computerized cognitive training program targeted at inhibition, as compared to a control condition. Inclusion criteria were: (i) >18 years old, (ii) willing to share name, date and place of birth, (iii) a PGSI-recent score ≥ 5 and (iv) being beneficiary of the French social security system and resident in France. The non-inclusion criterion was not speaking or understanding French. Eligible participants were included by phone. Participants were enrolled without any face-to-face interview through teleconsulting, they were located all over the French territory. They were then followed up online. This study met the French requirements for interventional studies and was approved by CPP/CNIL.

The study presented in the second part of the article is a nested study aiming to validate the French version of the GQoLS. Baseline data from the 98 included subjects were extracted.

#### 2.2.2. Measures

Baseline data were collected by phone: sociodemographic data (age, sex, employment); MINI 5.0 current major depressive episode; Alcohol Use Disorders Test (Audit-C); tobacco consumption; gambling characteristics (Problem Gambling Severity Index score (PGSI score), onset of loss of control of gambling); types of games (1. draw games, scratch cards or interactive online; 2. online sports betting; 3. online horse racing betting; 4. online poker; 5. online slot machines or other online casino games; 6. other games online) and offline gambling in the past 12 months (casino, circle games or racecourse); ratio of online/offline gambling; last 4-week time and money spent gambling; EuroQol five dimensions questionnaire (EQ-5D); and Gambling Quality of Life Scale (GQoLS). 

#### 2.2.3. Statistical Analysis

Descriptive analysis of the population demographics and disease severity was performed.

##### Distribution and Quality of GQoLS Items

According to the COSMIN checklist [18], we report structural validity, construct validity, internal consistency and hypothesis testing. 

The total score was obtained by summing all items. The GQoLS total score and item scores were described in the study population. A descriptive analysis of items explored their distribution. Floor effects were considered if more than 15% of respondents achieved the lowest possible score [18,19]. These items were removed. As the study population has a severe gambling disorder, it was expected that some items may be rated the highest response, thus we did not consider ceiling effects. Redundancy was considered when the estimation of Pearson’s correlation coefficient between items two by two was more than 0.70 [20], in which case the least adequate item was removed after discussion among the authors (AL and NB). 

##### Validity

Structural Validity

Only patients having completed all 50 items of GQoLS were included in the structural validity analysis. There was no data imputation. GQoLS is based on a reflective model. Exploratory factor analysis was conducted to explore the dimensional structure of the GQoLS. A scree plot of the correlation matrix of the remaining items was drawn; the number of dimensions of the scale was determined by Kaiser’s criterion (eigenvalues > 1) [21] and the Horn method (random simulations of data sets) [22]. A classical exploratory factor analysis using the most commonly used orthogonal rotation procedure (Varimax) was performed to define the GQoLS structure. Items were attributed to the dimension for which they had the highest loads; exceptions were: (1) when two loads were close, where the *a priori* dimension could be retained, or (2) when several loads were close, where the item could be attributed to the most relevant dimension. Items with loadings below 0.40 were removed from subsequent analyses and the exploratory factor analysis was repeated [23]. 

Construct Validity and Hypothesis Testing

External validity was assessed by comparing the GQoLS to the EQ-5D scale and EQ-5D VAS scale. Construct validity was assessed by Spearman correlations between the GQoLS scores and EQ-5D dimension scores (i.e., item level scores) and EQ-5D health state scores. Increased GQoLS or EQ-5D scores reflect a worse QoL, thus we hypothesized a positive correlation between GQoLS total score and each EQ-5D item score. We expected a moderate correlation between GQoLS total score and activity and anxiety scores, but a lower correlation between GQoLS total score and mobility, self-care, and pain/discomfort of EQ-5D. Higher EQ-5D health state visual analog scale (VAS) score reflects a better QoL, therefore we expected a negative and moderate correlation between GQoLS total score and EQ-5D VAS. 

External validity was also assessed by comparing the GQoLS to PGSI score. We hypothesized a positive correlation between GQoLS total score and PGSI score.

##### Internal Consistency

Internal consistency was assessed for the total score and GQoLS dimensions using Cronbach’s coefficient alpha. 

##### Item–Dimension Correlation

Item–dimension correlation was computed, omitting the item from its dimension, to avoid artificially inflated correlation (package psychometric; R).

All analyses were performed using R 3.0.3 software.

### 2.3. Development of a Brief Version of GQoLS

We also developed a short version of GQoLS (GQoLS-Bref) from final version of GQoLS. Two items were selected from each dimension according to three criteria [23]: those with the best internal consistency; those with the best loads in the factorial analysis, and thus the most associated with their dimension; and those with the best item–dimension correlations. Then, we kept the items that best met these three criteria and that were relevant and representative of their dimension in order to have a good content validity [24]. 

The structural validity, internal consistency and correlation with GQoLS were assessed according to the same methodology as for GQoLS.

#### Ethics

These studies were approved by the ethics committees (The Institutional Review Board of the Comité de Protection des Personnes-C.P.P IDF VII) on 3.9.16 and the C.P.P Sud Ouest et Outre Mer III on 29 November 2017. All subjects provided written informed consent. Confidentiality was preserved.

## 3. Results

### 3.1. Patient Sample

Ninety-eight participants with problem gambling were included. The sample was composed of 77 (79%) men and 21 (21%) women, with a mean age at the interview of 43.1 (±12.6) years. The mean PGSI score was 15.3 (±5.0) and the mean duration of the gambling disorder was 9.5 years (±7.6). Participants reported an equal proportion of online/offline gambling. The majority of the sample had not sought care. These data are summarized in Table 2.

#### 3.1.1. GQoLS Score Distribution

Only six subjects did not fully complete the questionnaire, giving a participation rate of 93.9% (Table 3). The GQoLS total mean score was 69.9 (±29.1), with a range of 0–121 for a possible score between 0–150. The percentage of missing data was between 2.0 and 5.1% for each item. The total range of responses was used for each item. Twenty-six items showed floor effect and were removed. Three pairs of items showed redundancy via a correlation greater than 0.7: items 40 § 41, 40 § 46 and 36 § 37. Items 40 and 36 were removed. The final version of the GQoLS contained 22 items.

##### Validity

Structural Validity

The structural validity was explored in two steps. These two steps are based on exploratory analysis. First, a scree plot showed a substantive principal dimension thus indicating the appropriateness of a total GQoLS score summing the items (Figure 1).

When considering the Kaiser–Guttman rule, five dimensions could be graphically identified (Figure 1 and Table 4) [20].

Secondly, we performed a classical exploratory factor analysis to define the GQOLS structure. The 5-factor structure was retained following classical exploratory factor analyses (Table 5). Item 14—Chasing had a loading < 0.4 and was removed. This 5-factor structure without item 14 explained 57% of the variance. Three a priori dimensions were partially confirmed. First, the negative emotion dimension was partially confirmed and corresponds to Factor 1. Factor 1 groups all items of the a priori “negative emotions” dimension and item 37—Not feeling oneself because of gambling of the a priori “preoccupation” dimension and item 39—Suffering from not understanding one’s own gambling behavior of the a priori “feeling of incomprehension” dimension. Secondly, the financial pressure dimension corresponded to Factor 2 which gathers all items of the a priori “financial pressure” dimension and two items of the a priori “loneliness” dimension (item 2—Relationships and item 4—Hiding distress). Factor 2 concerns lifestyle changes due to financial problems. Finally, the loneliness dimension was also partially confirmed and corresponds to Factor 3 which consists of three items of the a priori “loneliness” dimension: item 6—Feeling alone in the face of gambling, item 7—Feeling bad because of craving and item 8—Difficulty not to gamble when alone. Factor 4 groups two items of the a priori “relationships deterioration” dimension (item 22—Lying about gambling and item 27—Hiding gambling problems from relatives) and one item of the a priori “avoidance of helping relationship” dimension (item 51—Difficulty discussing my gambling problem with relatives). This factor refers to keeping the taboo, possibly due to the stigmatization of gambling. Factor 5 groups two items of the a priori “preoccupation” dimension: item 31—Gambling thoughts and item 32—Concentration.

The final GQoLS was composed of 21 items. The GQoLS total mean score was 38.3 (±13.6), with a range of 0–58 for a maximum possible score of 0–63. 

Construct Validity

Regarding EQ-5D, low-to-moderate and positive correlations were found between GQoLS and EQ-Anxiety. Low-to-moderate and negative correlations were found between GQoLS and EQ-Mobility. EQ-5D VAS was moderately correlated with GQoLS. PGSI was moderately-to-strongly correlated with GQoLS (Appendix A).

Internal Consistency

Cronbach’s alpha coefficient was 0.92, showing a high consistency. Cronbach’s coefficients for the five a posteriori dimensions ranged from 0.71 to 0.86.

Item–dimension correlations

Item–dimension correlations ranged from 0.46 (item 10–Factor 4) to 0.73 (items 16 and 17–Factor 1) (Appendix A).

#### 3.1.2. GQoLS-Bref

The selected items for each criterion are detailed in the supplementary section (Appendix A). For dimension 1—“Emotion”, we kept items 41 and 44, which were the most selected and representative of this dimension. Items 11 and 19 were retained for dimension 2—Lifestyle”, and item 7 for dimension 3—Loneliness”, via selection according to all three criteria, and were representative of their dimensions. For dimension 4—Taboo”, we kept item 27 because it showed better results than the other item in two of the three selection criteria and was similar to the other item regarding internal consistency. For dimension 5—“Preoccupation”, we chose item 31 because it had better results than the other item for the loading criteria and was equal for the other two criteria and the representativeness of the dimension.

Finally, GQoLS-BREF consisted of items 7, 11, 19, 27, 31, 41 and 44. A scree plot showed a unique dimension (data non-shown) thus indicating the appropriateness of a total GQoLS-BREF score summing the items. No other dimensions emerged applying the Kaiser–Guttman. The internal consistency was 0.79. The Spearman correlation between GQoLS and GQoLS-BREF was 0.97.

## 4. Discussion

The aim of this study was to develop and document the psychometric properties of the GQoLS, the first gambling-specific HRQoL instrument. We found a 5-dimension structure of GQoLS, with 21 acceptable items, in a population of French patients with current gambling disorders. The five *a posteriori* dimensions: “emotion”, “lifestyle”, “loneliness”, “taboo” and “preoccupation” were derived from the patient’s perspective.

The draft 50-item GQoLS was structured in seven a priori dimensions: loneliness, financial pressure, relationship deterioration, feeling of incomprehension, preoccupation with gambling, negative emotions, and avoidance of helping relationships [17]. An initially large number of items were kept exhaustively, anticipating a reduction during validation of items regarding a minority of very severe gamblers or illustrating particular contexts. The cognitive debriefing interviews showed excellent face and content validity.

In the psychometric property analysis of the GQoLS, the questionnaire was reduced to 21 items after eliminating items with floor effect, redundant questions or those not sufficiently correlated to a dimension. The low percentage of missing data supports good acceptability of the instrument among subjects with gambling disorders.

Structural validity found a substantial main dimension that allowed summing all the items and confirmed our hypothesis that GQoLS measures a single variable. Four other minor dimensions were found. The “emotion” and “lifestyle” dimensions were already explored in general QoL instruments. Indeed, the “lifestyle” dimension incorporates financial impact and relationship deterioration due to financial problems related to gambling. These domains are explored by the Short Form Health Survey (SF) scales, Quality of Life Inventory (QOLI) and Quality of Life Enjoyment and Satisfaction Questionnaire (Q-LES-Q) [14,25,26]. The “emotion” dimension was poorly explored in general QoL scales such as the World Health Organization Quality of Life (WHOQOL)-BREF (item 26: “How often do you have negative feelings such as blue mood, despair, anxiety, depression?”) [12]. However, the PRO development of GQoLS allows a more precise assessment of the emotions affected by gambling addiction. The “loneliness”, “taboo” and “preoccupation” dimensions have not been explored in QoL instruments used in gambling research [14] or in previous studies exploring QoL in gambling [17].

“Preoccupation” emerged in the qualitative analysis and was confirmed in the psychometric study. “Preoccupation” is the fourth criterion for diagnosis of problem gambling in DSM-5—“often preoccupied with gambling” [27]. Our results support this DSM-5 criterion and highlight its subjective importance for gamblers.

“Loneliness” was an *a priori* dimension emerging from the content analysis of the focus groups [17]. Although “loneliness” might be considered a negative emotion, the importance of this construct for participants during the qualitative analysis justified assigning it its own dimension [17]. This was confirmed by the factor analysis. “Loneliness” is separate from social isolation or solitude; it is a negative feeling that is a universal human experience arising from a discrepancy between the perceived and desired social connections [28,29]. Little is known about the relation between loneliness and addiction [29,30], particularly between gambling and loneliness. It is likely that the relationship between loneliness and addiction is reciprocal [31]. In substance use disorders studies, loneliness prevalence rates range from 35% to 79%, with 69% reporting this to be a serious concern [31,32]. “Loneliness” is experienced as problematic among people with substance use problems [30]. The PRO development of GQoLS has allowed the emergence of this little-known dimension that needs to be explored and considered in patient care.

Finally, “taboo” was found in structural validity as an HRQoL dimension in gambling disorders. This dimension is made up of items about avoidance of helping relationships and hiding and lying about one’s gambling problem. Problem gambling is characterized by high stigma [33], as problem gamblers are viewed to be more responsible for their difficulties than other addicts [34]. Moreover, stigma is a barrier to help-seeking in problem gambling [35] and could encourage keeping the taboo on gambling disorders. Keeping the taboo could lower connectedness. Low connectedness was demonstrated to mediate the relationship between depression and gambling disorders [36]. The development of GQoLS underlines the importance of considering taboo, and dealing with its potential sources, such as stigma, and will make it possible to measure the impact of interventions on this dimension. 

The positive correlation between GQoLS and the anxiety item of the EQ-5D, PGSI score and negative correlation with EQ-5D VAS supported the construct validity of the GQoLS. A moderate-to-strong positive correlation was found between GQoLS and PGSI score. This result is in line with previous studies that reported a correlation between HRQoL and the severity of the gambling disorder [37]. GQoLS and EQ-5D VAS are low-to-moderately correlated. GQoLS and EQ-5D are not superimposed. These differences probably arise via the PRO method used to develop the GQoLS. Finally, the GQoLS shows excellent internal consistency and good item–dimension correlations.

Our study has several limitations. First, the number of subjects included in this psychometric study is not optimal. It is customary to use five to ten participants per item, i.e., 100 to 200 subjects here [38]; however, this custom does not stand up to numerical simulations. For an instrument with five dimensions or less, the optimal number of subjects would be 300 [38]. However, we found good quality psychometric results with 98 participants. The second limitation of this study is that it is cross-sectional and does not allow testing of the instrument’s responsiveness to change. Finally, the GQoLS was developed and validated in French. A translation and validation in English and other languages would allow for wider use of this new instrument.

## 5. Conclusions

The GQoLS is the first HRQoL instrument specific to patients with gambling disorders and developed from the patient’s perspective. The GQoLS presents good psychometric properties. This instrument can be used in clinical research to demonstrate the effectiveness of an intervention on outcomes that are relevant from the patient’s perspective. Future use of the GQoLS should enable further validation of its sensitivity to change.

## Figures and Tables

**Figure 1 ijerph-19-10806-f001:**
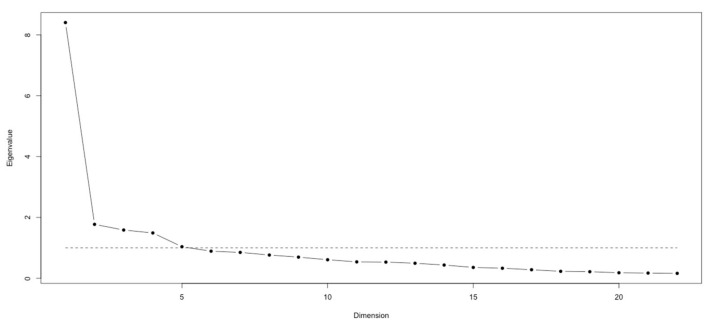
Scree plot. Legend: the dotted line represents the Kaiser rule.

**Table 1 ijerph-19-10806-t001:** Characteristics of cognitive debriefing interview patient sample.

*Characteristics*	Cognitive Debriefing Interview Sample (n = 11)
Age (years) (mean, sd)	43.6 (11.2)
Sex [n (%)]	
Male	9 (82)
Female	2 (18)
Relationship status [n (%)]	
Married or cohabiting	8 (73)
Single	3 (27)
Children [n (%)]	
Yes	6 (55)
No	5 (45)
Employment status [n (%)]	
Working	7 (64)
Not working	4 (36)
Financial difficulties [n (%)]	
Yes	7 (64)
No	4 (36)
Lifetime CPGI score (mean, sd)	20.3 (2.5)
Previous 12 months CPGI score (mean, sd)	12.6 (8.3)
Age of first gambling experience (years)(mean, sd)	17.1 (6.9)
Age at the beginning of gambling problem (years)(mean, sd)	30.1 (12.9)
Time since the beginning of gambling problem (years) (mean, sd)	13.5 (10.3)
Gambling frequency n (%)]	
No gambling session	6 (55)
<3 gambling sessions a week	1 (9)
≥3 gambling sessions a week	4 (36)
Current gambler [n (%)]	
Yes	6 (55)
No	5 (45)
Months since first contact with health services (mean, sd)	49 (82)
Types of gambling [n (%)]	
Scratch cards	5 (45)
Lottery	5 (45)
Horse racing betting	5 (45)
Sports betting	3 (27)
Poker (excluding online)	1 (9)
Slot machines	1 (9)
Other casino gambling	1 (9)
Online horse racing betting	0
Online sports betting	1 (9)
Online poker	2 (18)
Other online gambling	1 (9)
Past substance use disorder (Yes) [n (%)]	
Alcohol	2 (18)
Opiate	0
Cocaine	1 (9)
Cannabis	2 (18)
Sedative	0
Tobacco	2 (18)

sd: Standard Deviation; CPGI: Canadian Problem Gambling Index.

**Table 2 ijerph-19-10806-t002:** Description of the study population.

	(n = 98)
Sex (n, %)	
Female	21 (21%)
Male	77 (79%)
Age (mean, sd)	43.1 (12.6)
Professional activity (n, %)	
Employed	63 (64%)
Retired	8 (8%)
Unemployed	23 (24%)
Student	3 (3%)
PGSI (mean, sd)	15.3 (5.0)
Did not seek care (n, %)	56 (57%)
Duration of gambling problem (years) (mean, sd)	9.5 (7.6)
Type of online gambling in the past 12 months (n, %)	
Online lottery	26 (27%)
Online scratch cards	24 (25%)
Online horse racing betting	23 (23%)
Online sports betting	39 (40%)
Online poker	26 (27%)
Online slot machines and other online casino gambling	26 (27%)
Other online gambling	3 (3%)
None	28 (29%)
Type of offline gambling in the past twelve months (n, %)	
Lottery	66 (67%)
Scratch cards	73 (75%)
Horse racing betting	43 (44%)
Sports betting	42 (43%)
Poker	13 (13%)
Slot machines and other casino gambling	49 (50%)
Other gambling	0
None	4 (4%)
Online: offline gambling (n, %)	
Offline < Online	44 (45%)
Online > Offline	54 (55%)
Total loss in the last 28 days (in euros)	
Mean (sd)	975.6 (1818.5)
Min	−3984.0
Max	9996.0
Time gambled per day (hours)	
Mean (sd)	1.3 (1.5)
Min	0
Max	5.5
AUDIT-C (mean, sd)	4.1 (2.5)
Female	3.1 (2.3)
Male	4.3 (2.5)
Tobacco (n, %)	
No	55 (56%)
Yes	43 (44%)
Current major depressive disorder (MINI)	
No	95 (97%)
Yes	3 (3%)
EQ-5D VAS score (mean, sd)	68.7 (20.5)

**Table 3 ijerph-19-10806-t003:** Items and total score of the GQoLS distribution.

*Item Number*	*Percentage of Missing Data*	*Response Option (n (%))*		*Mean (SD)*
		Floor Effect	Ceiling Effect	
Item 1	2.0	17 (17.0)	9 (9.2)	1.34 (0.9)
Item 2	3.1	11 (11.2)	10 (10.2)	1.50 (0.8)
Item 3	2.0	22 (22.5)	6 (6.1)	1.22 (0.9)
Item 4	2.0	9 (9)	24 (24.5)	1.83 (0.9)
Item 5	3.1	29 (29.6)	7 (7.1)	1.1 (0.9)
Item 6	2.1	5 (5.1)	25 (25.5)	1.9 (0.9)
Item 7	2.0	10 (10.2)	30 (30.6)	1.9 (1.0)
Item 8	3.1	4 (4.1)	39 (39.8)	2.2 (0.9)
Item 9	4.1	47 (48.0)	13 (13.3)	0.9 (1.1)
Item 10	4.1	1 (1.0)	42 (43.0)	2.3 (0.7)
Item 11	4.1	12 (12.2)	25 (25.5)	1.8 (1.0)
Item 12	4.1	41 (41.2)	15 (15.3)	1.1 (1.0)
Item 13	4.1	74 (75.5)	3 (3.1)	0.3 (0.7)
Item 14	4.1	0 (0)	45 (45.6)	2.4 (0.7)
Item 15	3.1	69 (70.4)	4 (4.1)	0.4 (0.8)
Item 16	4.1	15 (15.3)	32 (32.7)	1.8 (1.1)
Item 17	3.1	57 (58.2)	7 (7.1)	0.7 (1.0)
Item 18	4.1	50 (51.0)	8 (8.2)	0.8 (1.0)
Item 19	3.1	9 (9.2)	15 (15.3)	1.7 (0.9)
Item 20	4.1	62 (63.3)	5 (5.1)	0.5 (0.9)
Item 21	4.1	45 (49.6)	8 (8.2)	0.9 (1.0)
Item 22	4.1	14 (14.3)	15 (15.3)	1.5 (0.9)
Item 23	4.1	38 (38.8)	9 (91.2)	1.0 (1.0)
Item 24	5.1	25 (25.5)	11 (11.2)	1.2 (1.0)
Item 25	4.1	49 (50.0)	9 (9.2)	0.8 (1.0)
Item 26	4.1	12 (12.2)	34 (34.7)	1.9 (1.1)
Item 27	4.1	30 (30.6)	18 (18.3)	1.3 (1.1)
Item 28	4.1	21 (21.4)	19 (19.4)	1.5 (1.1)
Item 29	4.1	43 (43.4)	7 (7.1)	0.8 (0.9)
Item 30	4.1	4 (5.1)	31 (31.6)	2.1 (0.8)
Item 31	3.1	11 (11.2)	10 (10.2)	1.6 (0.8)
Item 32	3.1	16 (16.3)	16 (16.3)	1.6 (1.0)
Item 33	3.1	55 (56.1)	4 (4.8)	0.6 (0.8)
Item 34	4.1	44 (44.9)	4 (4.1)	0.8 (0.9)
Item 35	3.1	9 (9.2)	23 (23.6)	1.8 (0.9)
Item 36	3.1	9 (9.2)	25 (25.5)	1.8 (0.9)
Item 37	4.1	30 (30.7)	7 (7.1)	1.1 (0.9)
Item 38	3.1	6 (6.1)	31 (31.7)	1.9 (0.9)
Item 39	4.1	2 (2.0)	54 (55.1)	2.4 (0.8)
Item 40	4.1	11 (11.2)	47 (48.0)	2.1 (1.0)
Item 41	4.1	29 (29.6)	17 (17.3)	1.3 (1.1)
Item 42	3.1	16 (16.3)	10 (10.2)	1.4 (0.9)
Item 43	3.1	5 (5.1)	32 (32.7)	2.1 (0.9)
Item 44	4.1	6 (6.1)	38 (38.8)	2.1 (0.9)
Item 45	3.1	1 (1.0)	49 (50.0)	2.4 (0.8)
Item 46	4.1	38 (38.8)	2 (2.0)	0.8 (0.8)
Item 47	4.1	11 (11.2)	11 (11.2)	1.5 (0.9)
Item 48	3.1	21 (21.4)	7 (7.1)	1.2 (0.9)
Item 49	3.1	40 (40.8)	9 (9.2)	1.0 (1.0)
Item 50	4.1	11 (11.2)	35 (35.7)	1.9 (1.1)
GQoLS total score	6.1	NA	NA	69.9 (29.1)

**Table 4 ijerph-19-10806-t004:** Eigenvalues of the correlation matrix.

*Dimension*	*Eigenvalues*	*Proportion of Variance*	*Cumulative Variance*
1	8.21	0.17	0.17
2	1.66	0.16	0.33
3	1.52	0.08	0.42
4	1.47	0.08	0.50
5	1.03	0.07	0.57
6	0.87	0.05	0.60

**Table 5 ijerph-19-10806-t005:** GQoLS 5-factor model (Varimax rotation–N = 92). Rotated Factor Pattern.

	*Factor 1* *EMOTION*	*Factor 2* *LIFESTYLE*	*Factor 3* *LONELINESS*	*Factor 4* *TABOO*	*Factor 5* *PREOCCUPATION*
2. Gambling has an impact on my relationships		0.58			
4. Hiding distress		0.56			
6. Feeling alone in the face of gambling	0.43		**0.40**		
7. Feeling bad because of craving			0.62		
8. Difficulty not gambling when alone			0.67		
10. Gambling expenses		0.65			
11. Financial difficulties		0.75			
16. Deprivations		0.65			
19. Sleep disturbance due to gambling-related financial problems		0.68			
22. Lying about gambling				0.42	
27. Hiding gambling problem from relatives				0.89	
31. Gambling thoughts					0.94
32. Concentration					0.43
37. Not feeling oneself because of gambling	**0.46**	0.45			
39. Suffering from not understanding one’s own gambling behavior	0.61				
41. Shame	0.73				
44. Anxiety	0.67				
45. Feeling loosy	0.68				
46. Angry	0.67				
48. Irritability	0.48				
50. Difficulty discussing my gambling problem with relatives				0.56	

Components < 0.4 were removed for clarity. Items with balanced loads on 2 factors were attributed to the more meaningful factor, in bold.

## Data Availability

The data in the article are from an ongoing study and the data cannot be shared at this time.

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
