# Peer review of "Development and Psychometric Properties of a New Patient-Reported Outcome Instrument of Health-Related Quality of Life Specific to Patients with Gambling Disorder: The Gambling Quality of Life Scale (GQoLS)"

_ijerph, 2022, doi:10.3390/ijerph191710806_

Round 1
Reviewer 1 Report
The paper describes the development and psychometric properties of a new instrument to measure the health-related quality of life of persons with gambling disorders. The introduction makes a good point for developing the scale, but given the focus on symptoms and consequences of gambling disorders, it would be better to describe these correlates better in the introduction. E.g. the study by Bonfils et al. is mentioned in the methods section, but it is not clear what the domains look like in detail. Also, given the similarities with other addictive disorders, it is worth looking into recovery and QoL (instruments) research in these populations and to refer to this literature in the introduction.
The authors make a clear distinction between QoL and HRQoL and it is important to make this distinction consistently throughout the paper, which is not always the case (e.g. ln 75, in the discussion). Also, the choice for HRQoL is associated with a specific conceptualisation of recovery as clinical recovery (limited to symptoms and health issues), while a broader definition of recovery (personal recovery) has become more prominent in addiction and mental health recovery.
The method is quite well described, except the first step (the construction of the 7 domains of problem gambling and its indicators, and how this led to the 94 items). Additional information is required here.
The cognitive debriefing phase appeared to be a separate phase, involving 11 respondents, but it is unclear how this relates to the third phase (validation study). Were these 11 participants also included for the psychometric analyses(n=98)? As recruitment took place at the same hospital, the recruitment strategy should be made clearer as well as how both phases related to each other.
The discussion is well elaborated and discusses the various domains of the GQoLS. The introduction made clear that the authors sought to develop a HRQoL-instrument, but this is not mentioned again in the discussion. Given the conceptual difference and HRQoL as frame of reference, this focus should be stated more clearly in the discussion. This also becomes clear in the items/domains retained in the final version. Research on addiction recovery makes clear that connectedness and belonging are crucial for (personal) recovery, which is partly covered in the questionnaire by the ‘loneliness’ domain, while the factor ‘environment’ has only one item that refers to this connectedness. It would be good if the authors could focus more on the relation between ‘connectedness’ and ‘QoL/recovery’ and other than health-related dimensions of recovery, as several authors have criticized the health-related approach of QoL (e.g. Cummins et al., 2004; Gomez et al., 2010; Van Hecke et al., 2018).
Minor remarks
Table 1 is structured in an odd way and the order of the items is uncommon. It would be more logical to start eg. with socio-demographic information and then report the other items
Under table 4, the word ‘secondly’ is used twice
Author Response
We thank very much the 2 reviewers and the editor for their time and encouraging comments.
Reviewer 1
1.The paper describes the development and psychometric properties of a new instrument to measure the health-related quality of life of persons with gambling disorders. The introduction makes a good point for developing the scale, but given the focus on symptoms and consequences of gambling disorders, it would be better to describe these correlates better in the introduction. E.g. the study by Bonfils et al. is mentioned in the methods section, but it is not clear what the domains look like in detail. Also, given the similarities with other addictive disorders, it is worth looking into recovery and QoL (instruments) research in these populations and to refer to this literature in the introduction.
R1. We thank the reviewer for this suggestion We developed the introduction with the following sentences:
“The areas of life explored by Qol and HRQoL instruments used in the past in gambling disorder were relationships with others, activities, physical state, psychological state, financial concerns, medical care and satisfaction with life. »
- The authors make a clear distinction between QoL and HRQoL and it is important to make this distinction consistently throughout the paper, which is not always the case (e.g. ln 75, in the discussion).
R2. We corrected accordingly.
- Also, the choice for HRQoL is associated with a specific conceptualisation of recovery as clinical recovery (limited to symptoms and health issues), while a broader definition of recovery (personal recovery) has become more prominent in addiction and mental health recovery.
R3. We thank the reviewer for these very interesting insights. We added the following sentence to detail our conceptual framework: “Our conception of HRQoL is then not QoL reduced to gambling, but all possible areas of life impacted by gambling disorder in a comprehensive and subjective meaning. An improvement in HRQoL in this meaning shall not necessary overlap with symptom disappearance, but more with personal recovery. »
- The method is quite well described, except the first step (the construction of the 7 domains of problem gambling and its indicators, and how this led to the 94 items). Additional information is required here.
R4. We gave more details on that the first step: “Seven key areas of HRQoL affected by problem gambling were identified by Bonfils et al. through a content analysis of 6 focus groups in 25 current or lifetime at risk problem gamblers, performed with ALCESTE software. The seven areas formed the basis of the hypothetical conceptual framework for the GQoLS” “NB, HJA and AL generated 94 items using the significant verbatim from the corpus of our previous study [15]. The items were generated using directly the verbatim of participants from the first study, as unprocessed as possible. Some items could then overlap, but we gave a particular attention to generate several items for each 7 areas»
- The cognitive debriefing phase appeared to be a separate phase, involving 11 respondents, but it is unclear how this relates to the third phase (validation study). Were these 11 participants also included for the psychometric analyses (n=98)? As recruitment took place at the same hospital, the recruitment strategy should be made clearer as well as how both phases related to each other.
R5. We explain further the recruitment and population: The 98 gamblers were recruited online only, no relation with the 2 hospitals were took place the first phase: “. Participants were enrolled without any face to face interview through teleconsulting, they were located all over the French territory. They were then followed up online.”
- The discussion is well elaborated and discusses the various domains of the GQoLS. The introduction made clear that the authors sought to develop a HRQoL-instrument, but this is not mentioned again in the discussion. Given the conceptual difference and HRQoL as frame of reference, this focus should be stated more clearly in the discussion.
R6. We added this precision in the first paragraph of the discussion to make it appear clearer. “The aim of this study was to develop and document psychometric properties of the GQoLS, the first gambling specific HRQoL instrument”
7.This also becomes clear in the items/domains retained in the final version. Research on addiction recovery makes clear that connectedness and belonging are crucial for (personal) recovery, which is partly covered in the questionnaire by the ‘loneliness’ domain, while the factor ‘environment’ has only one item that refers to this connectedness. It would be good if the authors could focus more on the relation between ‘connectedness’ and ‘QoL/recovery’ and other than health-related dimensions of recovery, as several authors have criticized the health-related approach of QoL (e.g. Cummins et al., 2004; Gomez et al., 2010; Van Hecke et al., 2018).
R7. See our response R3. Moreover, we added a paragraph in the discussion in reference to connectedness: “Keeping the taboo could lower connectedness. Low connectedness has been demonstrated to mediate the relation between depression and gambling disorder (Lee, Hyun, Choi, & Aquino, 2021). The development of GQoLS underlines the importance of considering taboo, dealing with its potential sources, like stigma, and will make it possible to measure the impact of interventions on this dimension. “
Minor remarks
- Table 1 is structured in an odd way and the order of the items is uncommon. It would be more logical to start eg. with socio-demographic information and then report the other items
R8. This is true, we re-organized the table 1
- Under table 4, the word ‘secondly’ is used twice
R9. I’m so sorry I don’t see it. Perhaps was it a misconstruction of the pdf? It seems correct on the word file .docx
Thanks, best regards
Reviewer 2 Report
I was glad to see the initiative for developing a new instrument in the field of gambling disorder. I would like to congratulate the authors for their work and I am looking forward to see how this instrument will progress and develop in the future, when implemented in a wider context and with different samples internationally.
I have one request for change and some suggestions to the authors.
Page 2, line 92: Please do not refer to the answer scale as a Likert scale. Likert scale is a scale of agreeing with a certain item (from totally disagree to totally agree). In the scientific community there is a tendency of mentioning the Likert scale for every multiple-point scale. Your scale is just a four-point scale of answers, and that is enough to mention.
Table 5: I have suggestions for the authors about the names of factors. They are just suggestions, as the authors decide about the names of the factors in their scale, but my opinion is that some names do not reflect the content very well.
Factor 2: my suggestion is to call it "Relationships and finances". Environment is a very broad term, and the content of all items in this factor seems to focus on these two elements.
Factor 4: my opinion is that "Stigma" is not the best name for this factor. We have different behaviours in this factor (lying, hiding, and difficulty of talking with others about gambling problems) - but we do not know the motives behind such behaviours. There might be stigma, but there might be many other reasons and underlying motives. Therefore, I suggest calling this factor "Concealing and disclosure" [or something across these terms].
Factor 5: since the whole scale is related to gambling, in my opinion, there is no need to specify that preoccupation is focused on gambling. Emotions (Factor 1), loneliness (Factor 3), and stigma (Factor 4) are all related to gambling, and it is not specified in the name of the factor. Accordingly, in my opinion, it is enough to call this factor just "Preoccupation".
All statistical procedures are done in line with scientific steps and standards, and descriptions are informative, clear, and coherent.
Author Response
We thank very much the 2 reviewers and the editor for their time and encouraging comments.
Reviewer 2
I was glad to see the initiative for developing a new instrument in the field of gambling disorder. I would like to congratulate the authors for their work and I am looking forward to see how this instrument will progress and develop in the future, when implemented in a wider context and with different samples internationally.
I have one request for change and some suggestions to the authors.
- Page 2, line 92: Please do not refer to the answer scale as a Likert scale. Likert scale is a scale of agreeing with a certain item (from totally disagree to totally agree). In the scientific community there is a tendency of mentioning the Likert scale for every multiple-point scale. Your scale is just a four-point scale of answers, and that is enough to mention.
R1: ok we deleted the term Likert
- Table 5: I have suggestions for the authors about the names of factors. They are just suggestions, as the authors decide about the names of the factors in their scale, but my opinion is that some names do not reflect the content very well. Factor 2: my suggestion is to call it "Relationships and finances". Environment is a very broad term, and the content of all items in this factor seems to focus on these two elements.
R2 . We appreciate this suggestion. We chose to change the factor label to LIFESTYLE
- Factor 4: my opinion is that "Stigma" is not the best name for this factor. We have different behaviours in this factor (lying, hiding, and difficulty of talking with others about gambling problems) - but we do not know the motives behind such behaviours. There might be stigma, but there might be many other reasons and underlying motives. Therefore, I suggest calling this factor "Concealing and disclosure" [or something across these terms].
R3. We appreciated the suggestion and changed the label to TABOO. We changed the text and discussion accordingly: “Moreover, stigma is a barrier to help-seeking in problem gambling [35] and could encourage keeping the taboo on gambling disorder »
- Factor 5: since the whole scale is related to gambling, in my opinion, there is no need to specify that preoccupation is focused on gambling. Emotions (Factor 1), loneliness (Factor 3), and stigma (Factor 4) are all related to gambling, and it is not specified in the name of the factor. Accordingly, in my opinion, it is enough to call this factor just "Preoccupation".
R4.Ok we deleted “with gambling"
All statistical procedures are done in line with scientific steps and standards, and descriptions are informative, clear, and coherent.
Thanks, best regards